# A patient–clinician James Lind Alliance partnership to identify research priorities for hyperemesis gravidarum

Caitlin R Dean ![ORCID] ,[1,2] Hyke Bierma,[2] Ria Clarke,[3] Brian Cleary,[4,5] Patricia Ellis,[6] Roger Gadsby,[1,7] Norah Gauw,[8] Karen Lodge,[1] Kimber MacGibbon,[9] Marian McBride,[4,10] Deirdre Munro,[4,11] Catherine Nelson-Piercy,[12] Margaret O'Hara,[1] Helen Penny,[13] Katherine Shorter,[14] René Spijker,[15] Jone Trovik ![ORCID] ,[16] Emma Watford,[1] Rebecca C Painter[2]

► Prepublication history and supplemental material for this paper is available online. To view these files, please visit the journal online (http://dx.doi.org/10.1136/bmjopen-2020-041254).

For numbered affiliations see end of article.

**Correspondence to**
Ms Caitlin R Dean;
c.r.dean@amsterdamumc.nl

## ABSTRACT

**Objective** There are many uncertainties surrounding the aetiology, treatment and sequelae of hyperemesis gravidarum (HG). Prioritising research questions could reduce research waste, helping researchers and funders direct attention to those questions which most urgently need addressing. The HG priority setting partnership (PSP) was established to identify and rank the top 25 priority research questions important to both patients and clinicians.

**Methods** Following the James Lind Alliance (JLA) methodology, an HG PSP steering group was established. Stakeholders representing patients, carers and multidisciplinary professionals completed an online survey to gather uncertainties. Eligible uncertainties related to HG. Uncertainties on nausea and vomiting of pregnancy and those on complementary treatments were not eligible. Questions were verified against the evidence. Two rounds of prioritisation included an online ranking survey and a 1-hour consensus workshop.

**Results** 1009 participants (938 patients/carers, 118 professionals with overlap between categories) submitted 2899 questions. Questions originated from participants in 26 different countries, and people from 32 countries took part in the first prioritisation stage. 66 unique questions emerged, which were evidence checked according to the agreed protocol. 65 true uncertainties were narrowed via an online ranking survey to 26 unranked uncertainties. The consensus workshop was attended by 19 international patients and clinicians who reached consensus on the top 10 questions for international researchers to address. More patients than professionals took part in the surveys but were equally distributed during the consensus workshop. Participants from low-income and middle-income countries noted that the priorities may be different in their settings.

**Conclusions** By following the JLA method, a prioritised list of uncertainties relevant to both HG patients and their clinicians has been identified which can inform the international HG research agenda, funders and policy-makers. While it is possible to conduct an international PSP, results from developed countries may not be as relevant in low-income and middle-income countries.

## Strengths and limitations of this study

► The research process was driven by patients and healthcare professionals with a specific interest in hyperemesis gravidarum.
► Participants from a wide range of countries participated in the first truly international James Lind Priority Setting Partnership.
► Results may be less relevant to researchers within developing nations where education and awareness may be more important priorities.
► There was crossover between stakeholder groups so that many healthcare professionals and carers were also patients, therefore healthcare professionals may have been under-represented.

## INTRODUCTION

Hyperemesis gravidarum (HG) is severe vomiting and nausea in pregnancy. Affecting between 1% and 3% of all pregnancies,[1] HG presents the major reason for hospital admissions in the first half of pregnancy. Yet, HG's pathophysiology has not been elucidated. Historically, HG has been an under-researched condition and the quality of HG research to date has been poor.[2 3]

A number of systematic reviews[2 3] have shown that there is insufficient evidence to provide a definitive recommendation on the best way to treat HG. These reviews also concluded that the overall quality of evidence for HG treatments was low, and that further research was therefore required.[3] The authors of the later review made several research recommendations in order of priority. However, neither of these systematic reviews engaged meaningfully with patients who had experienced HG. As a result, the research recommendations have been criticised by patient advocacy groups for this condition.[4 5]

All too often questions being addressed by the research community are not aligned with the priorities of patients or healthcare professionals who seek answers to guide practice and decision making.[6] In 2018, National Health Service England and the National Institute for Health Research (NIHR) conducted a needs assessment for medical research which highlighted that new research should only be undertaken where there is a defined evidence gap and a need which can be clearly articulated.[7] The James Lind Alliance (JLA) is a non-profit initiative which seeks to address this mismatch by bringing patients, carers and clinicians together into priority setting partnerships (PSP) to identify and prioritise the unanswered questions which they consider to be most important. Established in 2004, the JLA is funded by the NIHR and the PSPs to oversee the process with their established methodology and over 100 PSPs have been established since inception.[8]

The HG PSP came together following the second International Colloquium on Hyperemesis Gravidarum (ICHG) held in the UK in October 2017. During this event, leading HG researchers as well as patients from around the world agreed that setting Research Priorities for the condition was itself a priority and motivation for such a project was strong. It was noted that many of the topics relevant to patients are completely absent from the current literature,[9] a mismatch noted by other JLA initiatives.[6] Following this event, a subgroup of speakers and delegates with established international links to others in the field agreed to pursue the project and formed the initial PSP.

The HG PSP sought to identify the unanswered questions about HG treatment from patient and clinical perspectives and then prioritise those that patients and clinicians agree are the most important.

The objectives of the HG PSP were to:
► Work internationally with patients and clinicians to identify uncertainties about the effects of HG treatments and management which have not yet been answered by existing research.
► Determine by international consensus a prioritised list of those uncertainties, to guide future research.
► To publicise the results of the PSP among researchers, research commissioning bodies and the general public in order to stimulate research in these areas.

## METHODS
### Patient and public involvement
People affected by HG, including patients, their carers and adult offspring were involved throughout this project from inception in line with the JLA methodology. The lead researcher, CRD, is a patient representative and along with the patient representatives on the steering group ensured patient and carers were equal partners throughout the process including decision making around design, piloting surveys, participation and interpretation of results. The online HG patient community has been instrumental in disseminating research and will

be utilised again to ensure broad public dissemination of the results from this project.

### Steering group
The HG PSP steering group was established to oversee the project in accordance with the JLA guidelines.[8] The steering group was chaired by a facilitator from JLA. Members consisted of both patients and clinicians from a range of professions involved in the care and treatment of HG, see online supplemental file 1 for a list of members.

Due to the international nature of the project meetings were held online. The project ran from the first steering group meeting on the 21 March 2018 and cumulated with the final steering group meeting on 21 November 2019. The steering group's tasks were to provide input and consensus on the protocol and the survey, defining the scope of HG symptoms eligible, as well as the scope of the outcomes eligible for inclusion. Furthermore, the steering group monitored the progress of the project, provided input on the interpretation of participant feedback, and monitored the quality of the conclusions. All steering group decisions were reached by consensus.

Steering group meetings required the participation of at least three of each stakeholder groups to proceed.

### Protocol, definitions, scope
The steering group developed and agreed on the study protocol (online supplemental file 2 and available at: http://www.jla.nihr.ac.uk/priority-setting-partnerships/ hyperemesis-gravidarum/) and formally adopted terms of reference, see figure 1. The steering group agreed that no formal definition of HG would be applied, such as hospital admission; people who considered themselves to have experienced HG (or their partners/offspring/ carers) would be welcome to participate. Participation would not be limited by time since experience of the condition. However, questions specifically regarding mild nausea and vomiting of pregnancy or 'morning sickness' would be considered out of scope. It was unanimously agreed that the only other criteria that would render questions to be out of scope were those regarding complementary and alternative therapies, such as homeopathy, acupuncture and herbal remedies including ginger.

### Collection of research questions: initial survey
Uncertainties were collected using an online survey (via SurveyMonkey), which was also made available on paper. The survey was piloted by the steering group members.

Online dissemination of the survey followed an agreed social media strategy using the various partner organisation's platforms. Those steering group members who distributed paper copies within clinics accepted responsibility for collecting completed surveys and returning them to the lead researcher. Posters advertising the survey with a weblink and QR code were produced and distributed to relevant clinical setting where either steering group members or other engaged stakeholders were able to display them. Healthcare professional organisations from

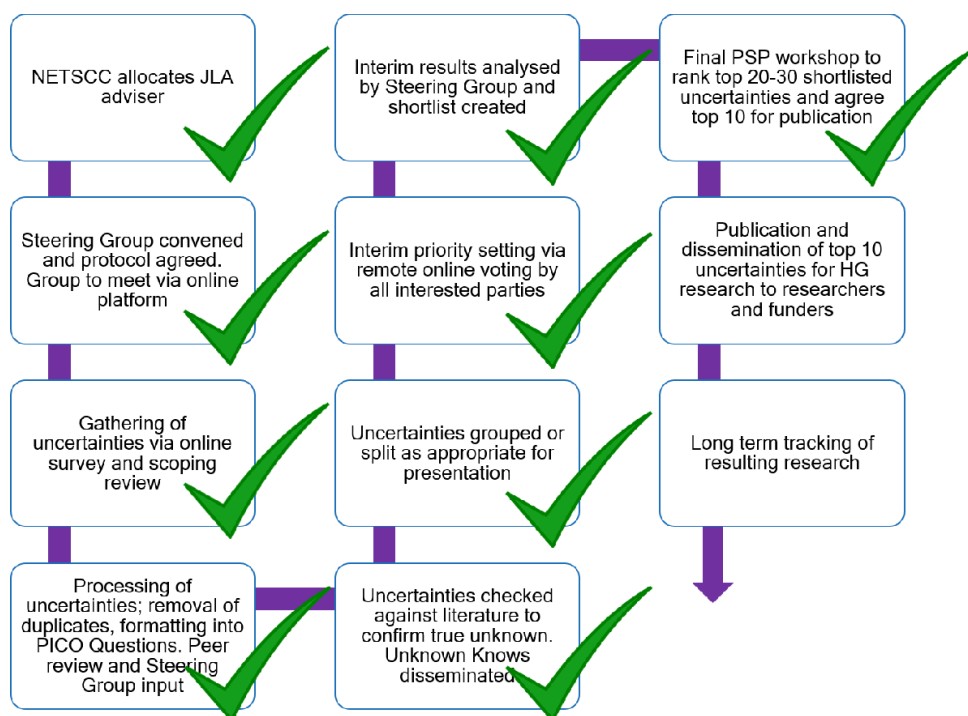

**Figure 1** The JLA process with steps completed to date. HG, hyperemesis gravidarum; JLA, James Lind Alliance; PICO, patient/problem, intervention, comparison, outcome; PSP, priority setting partnership; NETSCC, The NIHR Evaluation,Trials and Studies Coordinating Centre.

multiple countries were engaged with on social media platforms and directly to request assistance in promoting the survey to their audiences.

Participants could submit up to five questions regarding any aspect of HG research such as diagnosis, prognosis, treatment and the organisation of care. The survey platform could only be used once on any one device. The survey was translated into both Dutch and Norwegian and paper versions of both were made available online and preprinted. Information provided at the start of the survey about how and why to take part was also provided in video format (Viewable at: https://vimeo.com/285975872). The relevant ethics waivers were obtained for each hospital setting and a Health Research Authority certificate of non-research activity provided where required (online supplemental file 3).

After the first 2weeks of data collection, the demographics of respondents were reviewed by the steering group members, in order to allow advertising targeted at underrepresented stakeholder groups.

Participation in this stage of the project was not a prerequisite to participation in later stages such as the interim ranking and final workshop.

### Processing the data

Once the initial survey was closed, the raw data was downloaded. Foreign language versions on either paper or online were translated independently by both a healthcare professional and a patient representative for each language.

Respondents were allocated unique numbers and their raw questions separated into unique entries with the corresponding demographic information retained with each question.

The lead researcher thematically coded all responses, including out of scope questions (see online supplemental file 4) for codes). Queries were discussed within the steering group until consensus was reached. To ensure consistency of coding, the steering group checked the first 100 questions and, if this gave rise to significant disagreement, the number of questions to be checked could be expanded.

Indicative questions were formed which captured the meaning of the original questions in a logical and complete question. Each raw question was allocated to the indicative question which represented it. The steering group was divided into small groups to cross-check each raw question against the allocated indicative question to ensure that they were fully represented in lay language, and original meaning was not lost or skewed during the coding/question development process.

### Evidence checking

In order to identify the responses for which enough evidence of sufficient quality was available to consider them as answered that is, 'unknown knowns', an extensive check of the evidence was undertaken according to an agreed protocol (online supplemental file 5). Two reviewers, CRD and HB, labelled the search results according to the question references related to and CD

then cross checked each question against the evidence and the Royal College of Obstetricians and Gynaecologists (RCOG) and American College of Obstetricians and Gynaecologists (ACOG) guidelines for a suitably high level of evidence identified, that is, systematic review with conclusive results.

The steering group convened to discuss the evidence for each question. We considered a question sufficiently answered if a systematic review and meta-analysis in the last 10 years had provided a summary statistic with a narrow CI, which would make it unlikely to change with increased sample size. This was a deviation from the JLA recommendation that systematic review less than 3 years old are considered relevant and up to date,[8] however, the steering group agreed that this definition would be too narrow.

### Interim ranking survey

To narrow the long list down for the final workshop, so that a manageable number of questions, of around 25, could be prioritised at the workshop, an interim online survey was conducted. Participants were presented with an online survey (via SurveyMonkey) where they were asked to read each question individually and scored its importance on a scale of 1–10 where 1 was not at all important and 10 was the most important. Information was given at the start of the survey to encourage participants to use the full scale and to continue through the 65 questions. Several steering group members piloted the survey and comments were incorporated before launch. The questions appeared in a random order to each participant. The survey was only made available in English, however, it was distributed internationally. Partial completions were not excluded.

The survey was initially open from 19 August to 18 September 2019 when a review of the demographic data was conducted. Following the review, the survey remained open until the 30 September 2019 in order to increase representation of healthcare professionals and a recruitment drive on social media and via healthcare professional organisations was implemented.

### Consensus workshop

Potential participants for the final consensus workshop were invited via email. Specifically, participants of the earlier surveys who had requested further information on the project were emailed as well as registered delegates for the ICHG. Additionally, the workshop was advertised on social media through charity partner platforms and directly via email and word of mouth to steering group member contacts.

The workshop was facilitated by the JLA representative, the lead researcher (CRD) was present as an observer. Participants were divided into three groups and across two sessions were switched around so that the balance of groups was always evenly distributed with an equal mix of patients and professionals. The participants were presented with all 26 indicative questions, in conjunction

with the rank the indicative question had gained after the survey round. Healthcare, patient and carer ranks were separately presented to workshop participants. During the first two sessions the various groups ranked the questions 1–26 and the results were collated by the facilitators in order to reach an agreed consensus on the top 26.[8] The final session of the day saw the whole group convene, discuss any disputes regarding order, and make the final decision.

## RESULTS

### Uncertainty harvesting survey

The survey was open from 12 September to 16 November 2018, during which period two key healthcare professional conferences (general practice and early pregnancy events) were attended and paper copies were distributed among attendees.

A total of 1009 participants took part in the survey, although not all participants completed all the demographic questions (between 774 and 964/1009, depending on the question). Eligibility was not determined by completeness of demographic questions and those who did not complete demographic questions were not excluded. Of those who did answer any of the questions, 12.3% (n=118/963) were healthcare professionals and 97% (n=938/963) were patients and/or carers. Of the healthcare professionals who took part, 37% (n=44/118) had also suffered the condition themselves. Most participants identified as women (n=934). Only 30 people who identified as men took part of which 16 were healthcare professionals, 14 were carers (1 was both carer and professional), and one stated they had been diagnosed with HG. The UK and the Netherlands had the most engagement however questions were submitted from across the globe, see figure 2.

### Initial survey results

After removing blank and non-answers (n=19), a total of 2899 raw questions were received and coded. Of these, 37 questions were considered out of scope, 33 related to complementary therapies and 4 were out of scope for other reasons; 1 suggested a class action law suit on behalf of HG patients, two were requests to take part in research and one was excluded because no member of the steering group understood the abbreviations in the submitted question.

### Indicative questions

We developed 66 indicative questions, some of which had markedly more submissions than others, for example, 'what is the cause?' (n=282), 'which drugs are most effective?' (n=253) and 'what is the chance it will reoccur in a subsequent pregnancy?' (n=212). As the indicative questions were developed, the types of participants (patient, healthcare professional, carer, offspring) who asked each question were recorded and can be viewed in online supplemental file 6).

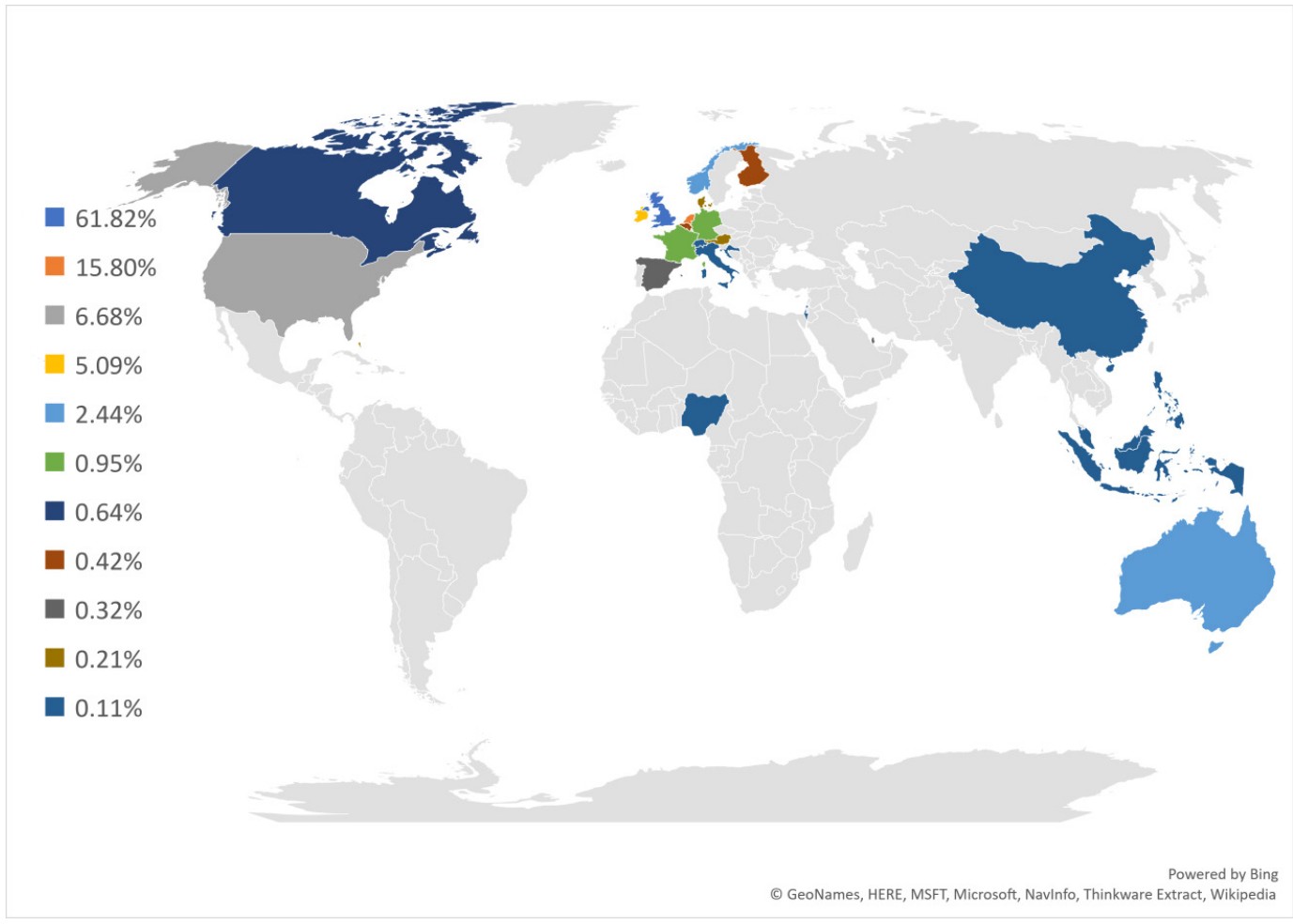

**Figure 2** Country of residence of participants in the initial survey.

### Evidence check

The search strategy revealed 627 references, which were labelled as to which question they related to by two reviewers. Each indicative question was then checked against the search results, the two identified guidelines (RCOG and ACOG) and the Cochrane Library, for whether it could be considered answered by systematic review.

The question 'does the sex of the fetus effect the occurrence/severity of the condition' was removed following the evidence check as it was deemed to be sufficiently answered by a systematic review; there is a slight increased ratio of female to male foetus in HG pregnancies (OR 1.27; 95% CI 1.21 to 1.34, based on x=2 672 040).[10] Five questions were considered to have partial answers, such as the role of plasma or urine ketoanalysis in diagnosis and management of HG,[11] none were considered to be sufficiently answered that no further research is required. In all 18 questions had no references related to them while one question, complications of HG, had 136 references the majority of which were case reports. The results of the evidence check with references per question is viewable in online supplemental file 6.

### Interim ranking survey

The interim survey contained all 65 indicative questions. 1115 participants took part in the ranking survey with 524 (47%) people ranking every question. Because the survey was displayed in a random order, all questions were ranked between 603 and 608 times each. Healthcare professionals represented 9% of participants and were particularly hard to engage with despite efforts to inform them about the survey and encourage participation. Carers accounted for 3.6% of participants and there were notable differences in their priorities to those of patients and healthcare professionals, for example, carers prioritised questions regarding symptom control for the sufferer over questions regarding harm to the fetus. International spread of participants was good with 32 countries represented.

The lack of healthcare professional representation was discussed with the steering group and a strategy to adjust for this was discussed with the JLA advisor. The question ranks were analysed and sorted by participant type. The first 21 questions were ranked above 25 by both patients and professionals and were automatically included. An additional 4, that healthcare professionals had ranked above these 21 where then included, to create a top 25.

However, this resulted in one question which previously been included by patient and healthcare professionals, and which was considered acutely important by the steering group without being sufficiently covered by other questions, to be eliminated. Therefore, 26 shortlisted questions went to the workshop for final ranking.

## Consensus workshop

Nineteen participants attended from nine countries; The Netherlands, UK, USA, Australia, Germany, Sweden, Nigeria, Ireland, Italy. There were 10 healthcare professionals and 9 patient representatives. There was good consensus between the groups and, in the final session, the whole group came together and agreed the top ten until there was unanimous agreement. The top 26 questions are shown in order of priority in table 1.

One participant from Nigeria expressed concern that the top ten reflected the priorities for developed, high-income nations and that in low-income and middle-income countries (LMIC) the priorities would be different to those reflected here. It was agreed that, despite the international nature of the project, the report would reflect that these priorities are not representative of LMIC priorities and the exercise would need to be repeated with participants representative of those countries.

## DISCUSSION

The top priorities for HG research, reached by stakeholder consensus, demonstrate the lack of knowledge about fundamental aspects of HG including effective treatment and prognosis. Our results can be used by researchers to identify future research priorities that are most relevant to patients, carers and healthcare professionals, as well as being of use to funders. The high numbers of patient responders to this PSP on HG, a transient, relatively rare condition limited to pregnancy, demonstrates the urgency felt among patients to prioritise progress in the identified research areas.

The theme of HG aetiology and consequences were both highly prioritised: We do not yet have sufficient knowledge about the pathophysiology of HG or the effects of the condition on the developing fetus such as exposure to malnutrition, chronic dehydration, heightened anxiety, maternal isolation and trauma. Such information is vital for patients and clinicians to be able to make evidence based informed decisions about when and how to treat. The importance placed on identifying a cause and cure for the condition contrasts to many other JLA PSPs for conditions in which aetiology and effective treatments have long been established. However, common to many previous PSPs, the HG priority top questions contain general uncertainties pertaining to prevention, best management and psychological impact on those affected.

While safety evidence regarding some medications is available and plentiful, particularly of the more commonly used medications such as doxylamine/pyridoxine, which is licensed for pregnancy sickness in various countries, and off-license treatments such as ondansetron,[12 13] questions remain over efficacy, combination therapy, dose, routes of administration and almost every other aspect of treating with these medications. The lack of robust information about the consequences of HG itself has led prescribers to assume that not treating is safer than treating.[14–16] In recent years, researchers have made a start identifying the risks of HG, including associations with autistic spectrum disorders among offspring,[17] cardiometabolic disturbances[18] and poor birth outcomes.[19] However, this is not sufficient to fully inform the important and difficult decisions required by patients and their healthcare providers during management of this condition.

While healthcare professional involvement was generally lower than patient involvement across JLA projects, healthcare professionals in this project only accounted for 12% and 9% of participants in the two survey rounds. This was disappointing given the efforts made to engage with healthcare professionals, the far-reaching connections from within the steering group and the presence of paper surveys in clinical settings in multiple countries. By comparison, the acne treatments PSP achieved 28% healthcare professional participation at the harvesting stage, and 64% at the ranking stage.[20] The liver glycogen storage disease PSP achieved 35% at both stages.[21] However, in comparison with other women's health and pregnancy conditions, engagement was similar. The miscarriage PSP had 1093 participants in round one, of which only 9.5% were healthcare professionals; the endometriosis PSP achieved 20% healthcare professional participation, of which 15% had experienced the condition themselves.[22] Likewise, as in this project, the cross over between healthcare professionals and patients was notable in other women's health conditions, such as 11% in the preterm birth PSP[23] and 31% in the pessary use for prolapse PSP.[24] The high level of patient engagement was anticipated by the authors as the patient population for this condition has previously demonstrated a strong interest for taking part in research with hundreds or even thousands of responses on surveys in a matter of days.[25 26] Although the low level of healthcare professional engagement was disappointing, it is not anticipated to negatively impact the goal of the PSP: to increase relevant research. Low healthcare professional engagement in pervious PSPs did not affect research stimulation: both 'sight loss' and 'tinnitus' PSPs have reported impressive funding stimulation and research uptake in the years following their PSPs[27 28] despite low healthcare professional engagement; 17% in the sight loss PSP[29] and 11%–19% in the various stages of the tinnitus PSP.[30] The lack of specialist healthcare professionals roles for certain conditions may account for some of the variation in engagement, for example, are no specific HG specialist nurse or midwife roles which may have affected engagement compared with other JLA projects such as diabetes, dermatology, and stroke which have nurse and physician specialists. Alternatively, the low healthcare professional

**Table 1** Top 26 ranked priorities for hyperemesis gravidarum research

| Ranking | Question |
| --- | --- |
| 1 | Can we find a cure? What novel or new treatments are being developed/tested/used elsewhere which could have a curative effect and to address all the symptoms of HG rather than just the vomiting? |
| 2 | How can we most effectively manage HG? What clinical support measure is most important to people who have had hyperemesis and what did they find most beneficial? For example, medical management, pharmaceutical review, nutrition support, rehydration, psychological support |
| 3 | What causes hyperemesis gravidarum? |
| 4 | Is HG preventable? What is the effect of preventative treatment or early intervention on the severity and duration of HG in a subsequent pregnancy? |
| 5 | What are the immediate and long-term effects of HG (including malnutrition and dehydration, stress) on the developing fetus (offspring)? |
| 6 | What are the immediate and long-term effects of the various medications/treatments on the developing fetus (offspring) throughout the various stages of pregnancy and in varying doses or combinations of treatments? |
| 7 | What are relative efficacies of the current medications and treatment options available? What is the optimal dose, route, timing and combination of the medications and what are the related side effects? |
| 8 | What are the immediate and long term, physical, mental and social consequences and complications of HG (including malnutrition and dehydration) on the pregnant person's body? (ie, metabolic impact, venous thromboembolism, depression, effects of dehydration) |
| 9 | What clinical measurements and markers are most useful in assessing, diagnosing, managing and monitoring hyperemesis? |
| 10 | What are the nutritional requirements of the first, second and third trimesters and how can people with HG achieve these goals? that is, oral supplements, fortifying food, dietary measures |
| 11 | How can symptoms of HG, other than vomiting, be effectively treated? For example, the nausea, excessive saliva, extreme sense of smell and fatigue. |
| 12 | Why are some cases of HG unresponsive to all antiemetics and how can we treat such cases? |
| 13 | What is the risk that HG will reoccur in a subsequent pregnancy? Does HG get progressively worse with subsequent pregnancies and what are the risk factors for reoccurrence? |
| 14 | Do clinical treatment guidelines for HG improve management and outcomes? And if so, how can guidelines be developed and implemented nationally (where none exist) and internationally for hospital and community settings? What should be included in guidelines? |
| 15 | How can people with a history of, or significant risk factors for HG be supported to plan for a pregnancy and does such planning improve outcomes? What should a pre-pregnancy plan contain? |
| 16 | How does HG impact on a person's (and their family's) quality of life? How does quality and efficacy of treatment impact that effect? |
| 17 | What is the currently level of knowledge about HG and its treatments among healthcare professionals (particularly GPs)? How can effective education for healthcare professionals be designed and delivered to improve the general knowledge and awareness of HG among healthcare professionals? |
| 18 | What is the most effective intravenous rehydration regimen; which solution in what quantity over what time period and how frequently? Does regular rehydration improve symptoms/outcomes/quality of life? |
| 19 | What is the effect of HG on mental health during (and after) pregnancy? What is the efficacy of psychotherapy on symptom management/pregnancy outcomes/quality of life? How can people access psychosupportive services during pregnancy? |
| 20 | What are the barriers to taking/prescribing medication for HG? How can the risk and benefits of HG and its treatments be better communicated to support informed decision making and consent to treatment? |
| 21 | What are the barriers to accessing treatments/services and how can we reduce them to improve access? |
| 22 | What healthcare services exist and how can they collaborate and be organised to better identify, treat and support people with HG? For example, do services such as outpatient clinics or intravenous at home, improve outcomes and reduce the physical/mental burden of the condition? |
| 23 | How can the condition be effectively managed in the community to prevent lengthy hospital admissions? |
| 24 | What self-management and coping strategies and treatments do people with HG find most helpful? |
| 25 | What employment rights do people with HG have and what financial support is available to them? |

Continued

| Table 1 | Continued |
|---|---|
| **Ranking** | **Question** |
| 26 | Do specific specialist healthcare professional roles for conditions such as HG improve outcomes? How can such roles be developed for midwives/nurses/doctors? |

HG, hyperemesis gravidarum.

engagement in the HG PSP may reflect a more general lack of interest in, or knowledge of HG among healthcare disciplines involved in HG care.

This PSP was the second fully international PSP so far and representation was obtained from a large number of countries. However, high-income countries were significantly over-represented and, despite efforts to engage doctors and patients from LMICs via social media and personal contacts, there was little engagement in the survey stages and only one representative at the final workshop. By comparison, the only other international PSP for liver glycogen storage disease had three representatives (27%) from LMIC (exclusively South American countries) at its workshop.[21] This is likely due to lack of funding although, for this project, travel bursaries were available including full bursaries for participants from LMICs. The JLA does not currently have a protocol for establishing how representative an international PSP should be or indeed how internationally relevant Priority Lists can be adopted in other countries. This can lead to significant research effort waste. For example, a PSP established a top 10 priority list for gestational diabetes in Canada in 2017 using JLA methodology,[31] yet a UK-specific group is conducting a JLA for diabetes in pregnancy this year.[32] Within our workshop, it was noted that the top 10 priorities may not be relevant for LMICs. While repeating this process in similar high-income countries would be considered wasteful, repeating the prioritisation process of the identified uncertainties in LMICs may be a worthwhile project.

## CONCLUSION

Following the JLA method, the international research arena can now direct resources to the top 10 priority research questions, which will have the greatest impact for people affected by HG and the healthcare professionals caring for them.

**Author affiliations**
[1]Pregnancy Sickness Support, Bodmin, UK
[2]Obstetrics and Gynecology, Amsterdam University Medical Centres, University of Amsterdam, Amsterdam, The Netherlands
[3]Obstetrics and Gynaecology, Frimley Park Hospital, Frimley, UK
[4]Hyperemesis Ireland, Dublin, Ireland
[5]School of Pharmacy, Royal College of Surgeons in Ireland, Dublin, Ireland
[6]James Lind Alliance, Southampton, UK
[7]Warwick Medical School, University of Warwick, Coventry, UK
[8]ZEHG Foundation, Amsterdam, Netherlands
[9]Hyperemesis Education and Research Foundation, Damascus, Oregon, USA
[10]Strategic Planning and Transformation, Health Service Executive, Dublin, Ireland
[11]Portiuncula University Hospital Galway, Galway, Ireland
[12]Obstetric Medicine, Guy's & St Thomas' Hospital, London, UK
[13]School of Psychology, Cardiff University, Cardiff, UK
[14]Early Pregnancy Unit, QMC, Nottingham University Hospital Trust, Nottingham, UK
[15]Medical Library, Amsterdam University Medical Centres, University of Amsterdam, Amsterdam, Netherlands
[16]Department Obstetrics and Gynaecology, Haukeland University Hospital, Bergen, Norway

**Contributors** CRD and RCP were the lead researchers, instigating the application to the JLA. RC organised the steering group meetings, designing and building the surveys, analysed the data, conducted the evidence check and wrote the majority of the final manuscript under supervision of RCP. PE (JLA Chair) chaired all the meetings, lead the consensus workshop and ensured compliance with methodology throughout. KL took minutes for steering group meetings, built and distributed surveys, organised the consensus workshop and acted as a facilitator in the workshop. RS conducted the searches for the evidence check and HB screened the results with CRD. Members of the steering committee RC, BC, RG, NG, KM, MM, DM, CN-P, MO, HP, KS, JT and EW all attended a majority of the meetings, agreed the initial protocol and the evidence check protocol, piloted and signed off the surveys and disseminated them, checked the raw questions against the indicative ones, reviewed the evidence check results and agreed the final longlist. CRD, RCP, KL, PE, RG, NG, KM, MM and MO were present at the consensus workshop. All authors reviewed and contributed to the final manuscript and approved it prior to submission.

**Funding** Funding was received from Pregnancy Sickness Support for the James Lind Alliance facilitator. The final workshop was funded by the Bikkja Trust who also provided some bursaries for attendance at the workshop.

**Map disclaimer** The depiction of boundaries on the map(s) in this article does not imply the expression of any opinion whatsoever on the part of BMJ (or any member of its group) concerning the legal status of any country, territory, jurisdiction or area or of its authorities. The map(s) are provided without any warranty of any kind, either express or implied.

**Competing interests** CRD is chair of the charity Pregnancy Sickness Support, which funded the JLA aspect of this project. RG and MO are trustees, KL is an employee and EW is a volunteer for Pregnancy Sickness Support. CRD and CN-P are trustees of the Bikkja Trust which funded the consensus workshop and provided some travel bursaries. CN-P has received consultancy and speaker fees from Alliance Pharma.

**Patient consent for publication** Not required.

**Provenance and peer review** Not commissioned; externally peer reviewed.

**Data availability statement** Data are available on reasonable request. All data relevant to the study are included in the article or uploaded as online supplemental information. All survey data and submitted questions will be available via the James Lind Alliance website without access restrictions (http://www.jla.nihr.ac.uk/priority-setting-partnerships/hyperemesis-gravidarum/).

**ORCID iDs**
Caitlin R Dean http://orcid.org/0000-0002-8812-5101
Jone Trovik http://orcid.org/0000-0002-3808-6407

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
