## [Reviewer comments · BMJ Open]

ARTICLE DETAILS

TITLE (PROVISIONAL)	A Patient-Clinician James Lind Alliance Partnership to Identify Research Priorities for Hyperemesis Gravidarum
AUTHORS	Dean, Caitlin; Bierma, Hyke; Clarke, Ria; Cleary, Brian; Ellis, Patricia; Gadsby, Roger; Gauw, Norah; Lodge, Karen; MacGibbon, Kimber; McBride, Marian; Munro, Deirdre; Nelson-Piercy, Catherine; O'Hara, Margaret; Penny, Helen; Shorter, Katherine; Spijker, René; Trovik, Jone; Watford, Emma; Painter, Rebecca

VERSION 1 – REVIEW

REVIEWER	Sandra Lowe RHW and University of New South Wales, Australia.
REVIEW RETURNED	19-Aug-2020

GENERAL COMMENTS	"We considered a question sufficiently answered if a systematic review and meta-analysis in the last 10 years had provided a summary statistic with a narrow confidence interval, which would make it unlikely to change with increased sample size. " This is a very narrow definition of "adequate evidence". " Ketoanalysis" ?plasma ?urine " to identify future research priorities that are most relevant to patients, carers and healthcare professionals, as well as being of use to funders." This study has met this intention
---

REVIEWER	Ellen Løkkegaard and Anne Ostenfeld Department of Obstetrics and Gynaecology Nordsjællands Hospital, Hillerød, University of Copenhagen, Institute of Clinical Medicine, Denmark
REVIEW RETURNED	27-Aug-2020

GENERAL COMMENTS	Thank you for this well conducted and very relevant study on research priorities in Hyperemesis Gravidarum (HG). Research priorities in this area are much needed and will hopefully influence future research for the benefit of people affected by HG. The study is relevant for several reasons as both high-quality research in HG is scarce and the etiology of HG is not clear, and evidence of treatment recommendations is low. Thus, the rationale for the paper is clear and important. The study is based on an international collaboration and includes information from both patients, carers and health care professionals. Thus, views of a broad specter of stakeholders are represented, which make the results highly relevant.
---

	We have minor comments for the authors: Page 4, line 35: Could you please be more specific on the definition of "complementary and alternative therapies" rendered out of scope? Page 5, line 57: Could you please include that the purpose of the Interim Ranking Survey was to narrow the 65 questions down to approximately 25 highest ranking questions? This is stated in the protocol, but when reading the manuscript, it is a little unclear how and why the 65 questions were reduced to 26. Page 6, line 50: Could you please elaborate on why "Missing data did not affect the... results"? Were sensitivity analyses performed? Page 6, line 58: Could you please specify the hyperemesis diagnose of the person who identified as male was diagnosed with. Was it hyperemesis gravidarum or hyperemesis of some other cause? Page 7, line 10: As mentioned before, could you be more specific on the definition of complementary therapies? Some might consider intravenous fluids complementary therapy yet question #18 is related to intravenous fluids. Page 9, line 3: Could you elaborate on whether it was preplanned that questions expressed by an underrepresented group of stakeholders would be weighed differently. If this was not preplanned, then please specify that you deviated from the protocol. Which we think in this case was indeed justifiable. Overall, the study had low participation from health care professionals. The participant recruitment process should be discussed. Could you have done something to improve the participation? For instance, invitations sent out to clinician networks might have included more countries. In general, the manuscript might be more reader friendly if abbreviations as JLA and PSP were written out.
--	--

VERSION 1 – AUTHOR RESPONSE

Point by point rebuttal

Reviewer 1 comments

Comment: This is a very narrow definition of “adequate evidence” (reference to sentence in our manuscript "We considered a question sufficiently answered if a systematic review and meta-analysis in the last 10 years had provided a summary statistic with a narrow confidence interval, which would make it unlikely to change with increased sample size. ")

Response: We agree that this is a narrow definition, however it was already a deviation for the James Lind methodology in order to expand their definition which limits suitable systematic reviews to the last 3 years. We felt further expansion would no longer be in line with JLA evidence guidelines. A line has been added, page 5 line 44-47 to clarify this.

Comment: " Ketoanalysis" ?plasma ?urine

Response: While urine ketoanalysis is commonly assessed in relation to hyperemesis the question refers to any form of ketoanalysis, plasma or urine without distinction. This detail has been added page 7 line 40.

Reviewer 2 comments

Comment: Could you please be more specific on the definition of "complementary and alternative therapies" rendered out of scope?

Response: We have clarified with an additional line that we are referring to non-medical based therapies such as homeopathy, acupuncture and herbal remedies including ginger. Page 4 line 25-26

Comment: Could you please include that the purpose of the Interim Ranking Survey was to narrow the 65 questions down to approximately 25 highest ranking questions? This is stated in the protocol, but when reading the manuscript, it is a little unclear how and why the 65 questions were reduced to 26.

Response: We agree this was not entirely clear. The interim ranking was performed in order to reduce the number of questions to be prioritised at the workshop to a manageable number and was conducted with online survey software. Additional information added at page 6 line 3-6 in order to clarify the rationale for the interim ranking survey and the method used.

Comment: Could you please elaborate on why "Missing data did not affect the... results"? Were sensitivity analyses performed?

Response: We agree the phrasing was confusing, and have rephrased the sentence. We were referring to the fact that we did not exclude survey input based on missing demographic data for individuals. The questions they submitted were still included in the results. The manuscript has been altered to clarify what was meant by this at page 7 line 2-4

Comment: Could you please specify the hyperemesis diagnose of the person who identified as male was diagnosed with. Was it hyperemesis gravidarum or hyperemesis of some other cause?

Response: Our data collection does not allow further comment or speculation on this, although the question they submitted demonstrated they had personal experience of hyperemesis gravidarum. The participant may have selected male in error or they may identify as male, we are not able to confirm or refute this due to the anonymous structure of the survey. We maintain that this result should be presented as it is although I have clarified the language, as highlighted at page 7 line 9-10.

Comment: As mentioned before, could you be more specific on the definition of complementary therapies? Some might consider intravenous fluids complementary therapy yet question #18 is related to intravenous fluids.

Response: As above, complementary therapies refers to non-medical interventions such as homeopathy and acupuncture. This has now been clarified in the methods page 4 line 25-26.

Comment: Page 9, line 3: Could you elaborate on whether it was preplanned that questions expressed by an underrepresented group of stakeholders would be weighed differently. If this was not preplanned, then please specify that you deviated from the protocol. Which we think in this case was indeed justifiable.

Response: Although it has not been pre-planned the JLA methodology and purpose of the steering group enables deviations from the protocol in unanticipated, dynamic situations, like this. This has been clarified in page 8 line 11-16

Comment: Overall, the study had low participation from health care professionals. The participant recruitment process should be discussed. Could you have done something to improve the participation? For instance, invitations sent out to clinician networks might have included more countries.

Response: Despite huge effort to engage international healthcare organisations internationally both on- and off-line recruitment was disappointing. This is discussed at length in page 10. Additional information on recruitment strategy has been added in the methods page 4 line 37-39 and page 6 line 16-8. Page 10 line 34-36 has also been added in the discussion to acknowledge this.

Comment: In general, the manuscript might be more reader friendly if abbreviations as JLA and PSP were written out.

Response: We have defined both these abbreviations at the start of the manuscript. We use JLA 26 times and PSP 36 times. While happy to alter the manuscript to write these out in full, this will

increase the word count by approximately 186. I therefore propose that this is an editorial decision and refer to the editor for advice.